# Attention Beats Concatenation for Conditioning Neural Fields

**Daniel Rebain**                                                  *drebain@google.com*
*University of British Columbia*
*Google Research*

**Mark J. Matthews**                                          *mjmatthews@google.com*
*Google Research*

**Kwang Moo Yi**                                                      *kmyi@cs.ubc.ca*
*University of British Columbia*

**Gopal Sharma**                                    *gopalsharma.research@gmail.com*
*University of British Columbia*

**Dmitry Lagun**                                                    *dlagun@google.com*
*Google Research*

**Andrea Tagliasacchi**                                              *taglia@google.com*
*Google Research*
*Simon Fraser University*

**Reviewed on OpenReview:** *https://openreview.net/forum?id=GzqdMrFQsE*

## Abstract

Neural fields model signals by mapping coordinate inputs to sampled values. They are becoming an increasingly important backbone architecture across many fields from vision and graphics to biology and astronomy. In this paper, we explore the differences between common conditioning mechanisms within these networks, an essential ingredient in shifting neural fields from memorization of signals to generalization, where the set of signals lying on a *manifold* is modelled jointly. In particular, we are interested in the scaling behaviour of these mechanisms to increasingly high-dimensional conditioning variables. As we show in our experiments, high-dimensional conditioning is key to modelling complex data distributions, thus it is important to determine what architecture choices best enable this when working on such problems. To this end, we run experiments modelling 2D, 3D, and 4D signals with neural fields, employing concatenation, hyper-network, and attention-based conditioning strategies – a necessary but laborious effort that has not been performed in the literature. We find that attention-based conditioning outperforms other approaches in a variety of settings.

## 1 Introduction

Neural fields, also called coordinate-based neural networks, have demonstrated excellent results in learning complex signals. Neural fields learn to reproduce signals by mapping input coordinates to output values. They are most commonly implemented as Multilayer Perceptrons (MLP) and have found widespread application over a range of tasks: 1D time sequences (Sitzmann et al., 2020), 2D image regression (Tancik et al., 2020), 3D radiance fields (Mildenhall et al., 2020), and 4D light fields (Suhail et al., 2022).

Many previous works overfit to a single example (e.g. an image, sound, object, or scene) (Sitzmann et al., 2020; Mildenhall et al., 2020), employing neural networks to "memorize" the target signal. However, there are many potential applications which require jointly representing *multiple* signals, commonly as a distribution over a

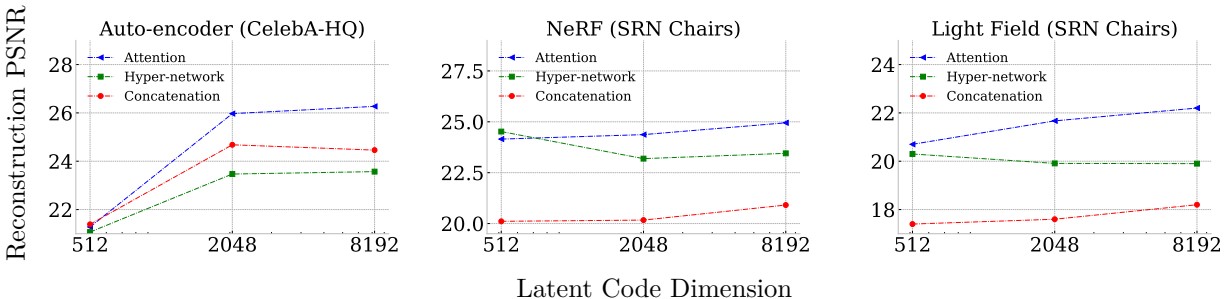

Figure 1: **Main results** − The relationship between representation effectiveness, as measured by PSNR, and representation complexity, as measured by latent code size; the plot reveals how attention conditioning outperform concatenation and hyper-networks when scaling to larger representations.

manifold (a smooth subspace of all possible signals). This generalization of neural fields has been achieved though various means: latent vector conditioned MLPs (Park et al., 2019), hyper-networks (Sitzmann et al., 2020), and set-latent transformer representations (Sajjadi et al., 2022). In all these means, the location of a particular signal on the output manifold is represented with some form of *latent code*.

The properties of the latent codes are important in determining what a conditional network is capable of modelling. Of particular interest to us is the *dimensionality* of the latent code, which directly determines the dimensionality of the manifold on which the output signals lie. If we wish to perfectly model some distribution of data, then we must use a latent dimension at least equal to that of the manifold on which that distribution lies. We demonstrate this effect on a toy dataset where we explicitly control the dimensionality of the data manifold, and find that increasing dimension of this manifold corresponds to worse reconstruction quality across different models.

Based on this observation, and inspired by methods such as VQ-VAE (van den Oord et al., 2017), which have utilized high-dimensional conditioning to great effect in CNN-based networks, we wish to determine whether neural field networks are capable of scaling up to condition on such high-dimensional latent codes, and what factors may affect this. Unlike CNNs, which "share" computation across the different (discrete) output samples of signals that they model, neural fields typically repeat the majority of their computation for each sample. As such, the question of how to most efficiently incorporate high-dimensional conditioning signals into a neural field network does not have an obvious answer – a problem which we intend to address in this paper. To facilitate this, we undertake a comprehensive study of three common methods of conditioning on latent inputs in neural fields: concatenated latent vectors, hyper-networks, and attention-based set latent representations. We evaluate each of these methods across three application domains: image auto-encoding, novel view synthesis using neural volumetric rendering, and light field networks.

Performing the experiments reported in this paper required a very large amount of compute time: on the order of 23$k$ GPU-hours. Due to the significant expense involved, we chose experimental parameters carefully and focused on architectural choices which appeared most likely to affect the outcome of the experiments, and therefore the conclusions of our analysis. We also did not run any additional hyper-parameter tuning or searches on top of our primary sweeps and ablation. It is our hope that by incurring this expense and sharing our conclusions, others can avoid the need to run similarly costly experiments when making design decisions for neural field implementations.

In summary, our contributions include:

- A series of extensive quantitative evaluations, where we compare how concatenation, hyper-network, and attention-based neural field architectures perform in modelling high-entropy data distributions for a number of application domains: 2D image auto-encoding, 3D radiance fields, and 4D light fields, benchmarked on standard datasets.
- An ablation study comparing different approaches to conditioning MLP neural field networks by concatenation when the latent dimension is very high. We find that for very large latent codes, splitting the code between the hidden layers of the MLP provides the best results;

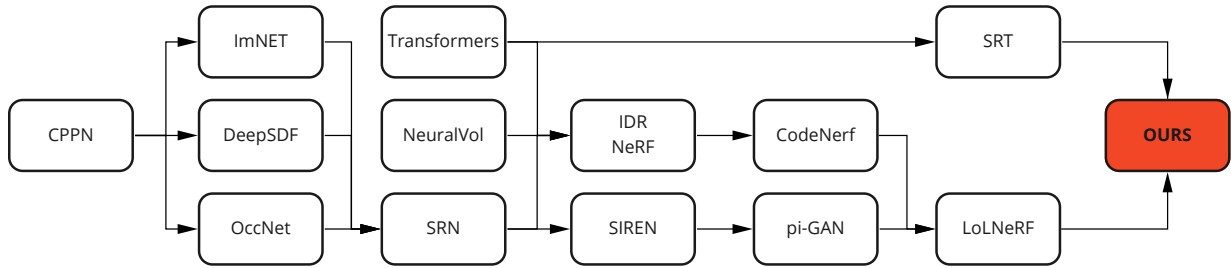

Figure 2: **Related works** − a temporal overview of how techniques have evolved, with arrows roughly denoting the dependency graph of technical contributions. At a high-level, we reveal how recently proposed coordinate network architectures (Sajjadi et al., 2022) are significantly more effective than MLPs (Stanley, 2007) for the task of conditional image generation (Rebain et al., 2022). Abbreviations for the various methods are highlighted in Section 2.

- An analysis showing that attention-based conditioning is broadly more effective for high-dimensional latent codes, given a particular compute budget.

## 2 Related Works

Our work mainly focuses on coordinate-based networks, also called neural fields. Introduced by Stanley (2007) under the name of Conditional Pattern Producing Networks (**CPPN**), recently became popular as differentiable representations for 3D data (Tewari et al., 2021; Xie et al., 2022). We now position our work with respect to the temporal progression of research in this topic (Figure 2), and in particular discussing how *conditioning* is implemented in architectures.

**Coordinate networks w/ 3D supervision**. Neural implicit networks for 3D shape representation receive as input a 3D point coordinate along with a latent encoding the shape and output either occupancy (**OccNet** (Mescheder et al., 2019) and **IMNet** (Chen & Zhang, 2019)) or (truncated) signed distance fields (**DeepSDF** Park et al. (2019)). These architectures are either trained with an auto-encoder (IMNet, OccNet) or an auto-decoder (DeepSDF) fashion, and rely on direct 3D supervision for training. Conditioning is implemented either by input concatenation (IMNet, DeepSDF), or by modulation of activations (DeepSDF).

**Coordinate networks w/ 2D supervision**. Moving away from 3D supervision, Scene Representation Networks (SRN) (Sitzmann et al., 2019) pioneered the use of 2D *images as supervision* of an underlying 3D neural scene, by relying on differentiable ray-marching. Similarly to previously discussed works, each scene in SRN is represented as latent code while conditioning realized via hyper-networks (Ha et al., 2017), but the generated images lack fine-grained details. Meanwhile, **NeuralVolumes** (Lombardi et al., 2019) introduced the use of *differentiable volume rendering* as a way to learn models with high visual-quality, but still relied on grids for scene representation. Marrying methods that store radiance within a coordinate network (Ren et al., 2013) with differentiable volume rendering (NeuralVolumes), neural scene representations (**SRN**), Mildenhall et al. (2020) introduced Neural Radiance Fields (**NeRF**) as an effective way to capture a 3D scene via differentiable rendering, and achieved novel-view synthesis results of unforeseen visual quality[1].

**Overcoming the spectral bias**. To represent high-frequency signals in neural scene representations one must overcome the so-called "spectral bias" of MLP architectures (Basri et al., 2020; Rahaman et al., 2019) – an inductive bias that causes fully connected neural networks to prefer representing low-frequency signals. NeRF overcame this issue by borrowing the ideas of (sinusoidal) *positional encoding* from transformers (Vaswani et al., 2017), SIREN (Sitzmann et al., 2020) proposed the use of sinusoidal activations (and careful initialization). These are relatively similar solutions, with the core difference that sinusoidal encoding is applied at the input

---

[1]Concurrent work by Yariv et al. (2020) and Niemeyer et al. (2020) is similar in spirit but have a more restrictive setup.

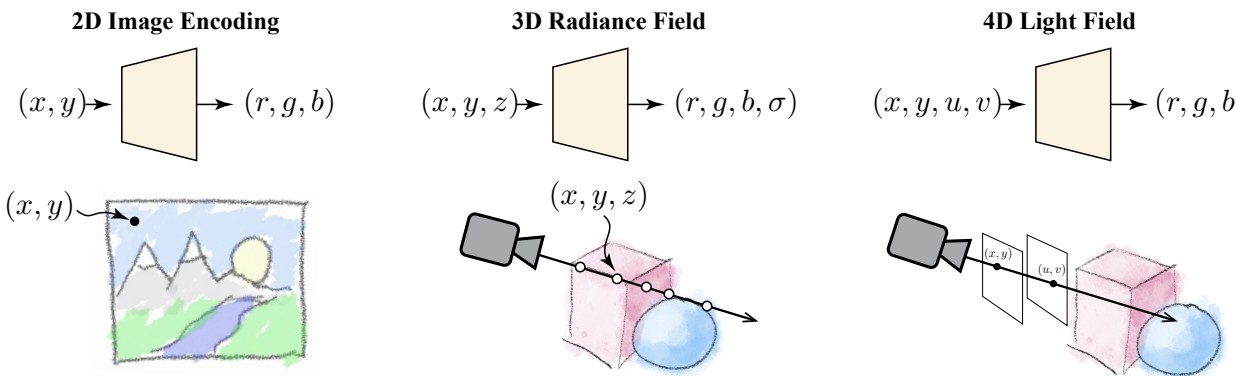

Figure 3: **Application domains** – We consider three main application domains: 2D image encoding, 3D radiance fields (radiance integrated along rays, but without directional dependence), 4D light fields.

layer in NeRF, and throughout the entire architecture in SIREN. Further, note that both of these solutions originally targeted the representation of a *single* scene, that is, working in the overfitting (i.e. non-conditional) training regime; the next section will cover multi-scene training, which is achieved via conditioning.

**Conditional networks w/ 2D supervision**. The recent **pi-GAN** (Chan et al., 2021) and **CodeN-eRF** (Jang & Agapito, 2021) demonstrate conditioning FiLM SIREN and MLP models respectively to obtain (class-specific) conditional neural radiance fields for novel view synthesis. While CodeNeRF struggles with low image quality, pi-GAN is capable of generating high quality samples by leveraging generative adversarial networks. Somewhat concurrently to CodeNeRF, Rebain et al. (2022) has shown that, in the conditional novel view synthesis setting, **LoLNeRF** can achieve competitive performance in comparison to GANs given a sufficiently large network and latent code. However, we reveal how there are diminishing returns in employing large MLP networks and large latent codes when MLP architectures are employed. We investigate this problem from the point of view of representation power. Our extensive experiments reveal how recently proposed alternatives to MLPs for coordinate-based neural networks (Sajjadi et al., 2022)(**SRT**) overcome this issue across a number of application domains.

## 3 Method

We experimentally analyze the representation power of different neural field conditioning mechanisms. With that objective, we first introduce the basics of neural fields in Section 3.1, their training strategies in Section 3.2, and their architecture in Section 3.3.

### 3.1 Neural Fields

Given a signal (1D for audio, 2D for images, 3D for geometry etc., See Figure 3) mapping a bounded set of coordinate inputs $\mathcal{X} \subset \mathbb{R}^d$ to outputs $\mathcal{O} \in \mathbb{R}^n$, a neural field $f_\theta$ with parameters $\theta$, represents a signal, taking a coordinate $\mathbf{x} \in \mathcal{X}$ as input and output $f_\theta \in \mathcal{O}$:

$$f_\theta : \mathbb{R}^d \to \mathbb{R}^n, \quad \mathbf{x} \mapsto f_\theta(\mathbf{x}) \tag{1}$$

Examples include fields receiving as input:

- $\mathbf{x} \in \mathbb{R}^1$ 1D time coordinates, outputting scalar audio intensity (Sitzmann et al., 2020);
- $\mathbf{x} \in \mathbb{R}^2$ 2D pixel coordinates, outputting RGB color for pixels (Stanley, 2007);
- $\mathbf{x} \in \mathbb{R}^3$ 3D coordinate, outputting a (truncated) signed distance from a shape (Park et al., 2019);
- $\mathbf{x} \in \mathbb{R}^4$ 4D encoded camera rays, outputting pixel color (Sitzmann et al., 2021);
- $\mathbf{x} \in \mathbb{R}^5$ 3D coordinate and 2D view direction, outputting density and radiance (Mildenhall et al., 2020).

Networks can be supervised minimizing a loss function over the observations:

$$\arg\min_{\theta} \mathop{\mathbb{E}}_{x \sim \mathcal{X}} \left[ ||f_\theta(\mathbf{x}) - S(\mathbf{x})||_2^2 \right]. \tag{2}$$

When using MLP architectures to implement $f_\theta$, directly inputting the coordinates to the neural network, they have a bias towards representing low-frequency signals (Rahaman et al., 2019), leading to low-quality signal reconstruction. To encourage high-frequency signal learning, the input coordinates may be independently mapped to higher dimension using sine and cosine functions with increasingly large frequencies (Vaswani et al., 2017; Mildenhall et al., 2020):

$$\gamma(x) = (\sin(2^0 \pi x), \sin(2^1 \pi x), ..., ..., \sin(2^{l-1} \pi x), \cos(2^0 \pi x), \cos(2^1 \pi x), ..., \cos(2^{l-1} \pi x)) \tag{3}$$

Several other formulations of position encoding have been explored (Rahimi & Recht, 2007; Barron et al., 2021; Tancik et al., 2020), but we use this one in our experiments as it has proven widely effective and remains commonly used.

### 3.2 Neural Fields: training methodology

Coordinate networks can be used to overfit (memorize) a bounded signal for such purposes as image denoising (Sitzmann et al., 2020) or novel view synthesis of scenes given a finite set of input images and associated poses (Mildenhall et al., 2020), but this memorization process does not generalize to new inputs. To overcome these limitations, conditional networks have been designed, where the decoder takes as input an *instance-specific* latent code $\mathbf{z} \in \mathbb{R}^m$ along with the coordinates to regress desired quantities. Below, we describe two approaches to training networks conditioned on latent codes: ① auto-encoders[2] and ② auto-decoders.

**Auto-encoder**. An auto-encoder (or encoder-decoder) employs a domain-specific encoder $\mathcal{E}$ parameterized by $\theta_e$ that takes a dataset element $\mathbf{I} \in \mathcal{I}$ as input and outputs the latent code as $\mathcal{E} : \mathbf{I} \mapsto \mathcal{E}(\mathbf{I}) = \mathbf{z} \in \mathbb{R}^m$. The element $\mathbf{I}$ can be an image, point cloud, audio signal etc. A decoder $\mathcal{D}$ parameterized by $\theta_d$ accepts the latent code and reconstructs the input as $\mathcal{D} : \mathbf{z} \mapsto \mathcal{D}(\mathbf{z}) = \hat{\mathbf{I}}$. These networks are trained to jointly optimize encoder and decoder parameters using the reconstruction objective:

$$\arg\min_{\theta_e, \theta_d} \mathop{\mathbb{E}}_{\mathbf{I} \sim \mathcal{I}} \left[ ||\mathcal{D}_{\theta_d}(\mathcal{E}_{\theta_e}(\mathbf{I})) - \mathbf{I}||_2^2 \right]. \tag{4}$$

**Auto-decoder**. Also known as Generative Latent Optimization (GLO) (Bojanowski et al., 2017), auto-decoders are a form of generative model which generates an output conditioned on a latent code $\mathbf{z}$. Here, the latent code is not predicted by an encoder, rather jointly optimized with the decoder parameters. A code-book $\mathcal{Z} = \{\mathbf{z}_i\}$ is used where each row consists of a latent code corresponding to each training instance. This alleviates the need of designing task-specific encoder. This neural network is trained to optimize the parameters of the decoder $\theta_d$ and the latent codes $\mathcal{Z}$ using the reconstruction loss:

$$\arg\min_{\mathcal{Z}, \theta_d} \mathop{\mathbb{E}}_{(\mathbf{I}, \mathbf{z}) \sim (\mathcal{I}, \mathcal{Z})} \left[ ||\mathcal{D}_{\theta_d}(\mathbf{z}) - \mathbf{I}||_2^2 + \rho(\mathbf{z}) \right], \tag{5}$$

where $\rho$ is an optional regularization function applied to the latent $\mathbf{z}$ when a particular distribution is desired. For example, choosing $\rho(\cdot) = \| \cdot \|_2^2$ results in the codes $\mathcal{Z}$ being distributed as a Gaussian (Park et al., 2019). During inference, commonly referred to as *test-time optimization* in the literature, the latent code is discovered by optimizing the above loss function while keeping the network parameters constant:

$$\arg\min_{\mathbf{z}} \mathop{\mathbb{E}}_{\mathbf{I} \sim \mathcal{I}} \left[ ||\mathcal{D}_{\theta_d}(\mathbf{z}) - \mathbf{I}||_2^2 + \rho(\mathbf{z}) \right], \tag{6}$$

---

[2]An extension of auto-encoders, Variational Auto-Encoders (VAE) (Kingma & Welling, 2013), are a probabilistic formulation that define a distribution of latent codes that generate elements matching the input dataset distribution. Generative Adversarial Networks (GAN) (Goodfellow et al., 2014) are another common method of synthesis that map a latent code to output. However, for the purpose of this paper we do not analyze VAEs and GANs, as we are not interested in generative aspects of the problem, and benchmark our results as a regression task.

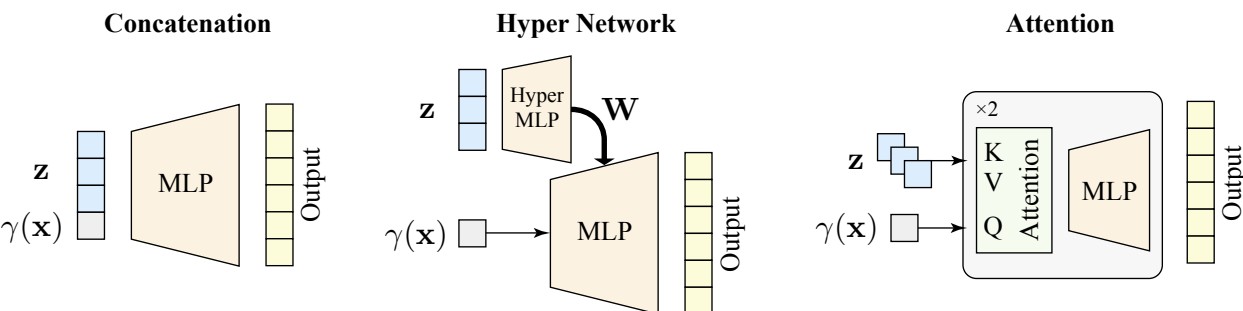

Figure 4: **Architectures** − Multi-layer perceptrons (MLPs) can be conditioned using a concatenation of coordinate and latent code (left). Hyper-networks use a secondary MLP that computes weights for the primary network which operates on the coordinate (middle). Attention networks condition on a set-latent representation that is incorporated into the network through attention layers, with query vectors being derived from the coordinates (right).

### 3.3 Neural Fields: decoder architectures

Many neural field architectures have been proposed in the literature. In this section, we review some common varieties including the ones tested in our experiments. Of particular relevance is the different ways in which these networks incorporate *conditioning* signals. For additional details on how we implement these architectures, see Section 4 and the Appendix.

**Concatenation** − **Figure 4 (left)**. Arguably the simplest way to construct a neural field is with an MLP applied to the input coordinates (either as $\mathbf{x}$ or $\gamma(\mathbf{x})$), and the simplest method to condition it by *concatenating* a latent vector to these coordinates (Park et al., 2021; Martin-Brualla et al., 2021). This input is commonly re-concatenated to one or more hidden layers using *skip connections* to improve stability (Chen & Zhang, 2019). This approach has proven effective for many applications, but has undesirable characteristics such as requiring $O(k(m + k))$ parameters for the linear layers, where $m$ is the latent dimension, and $k$ is the hidden layer width. Critically for our purposes, when $m$ becomes larger than $k$, the hidden layers effectively perform a linear dimensionality reduction of the latent codes, limiting the network's ability to utilize high-dimensional latents effectively. This can be partially mitigated by partitioning the latents to be distributed among the hidden layers, but even this is limited by the layer count; see Section 4.3.

**Hyper-networks** − **Figure 4 (middle)**. One approach to overcoming the inefficiency of concatenation in MLPs is hyper-networks (Ha et al., 2017; Sitzmann et al., 2019). In this approach, a secondary network $\Psi(\mathbf{z})$ applied to the latent code $\mathbf{z}$ regresses the *weights* $\phi$ of the MLP $f_\phi$, possibly allowing for a more efficient incorporation of latent information. While the coordinate network itself is very efficient in this setup, it off-loads complexity to the hyper-network, which can become very large in order to predict entire layer weights as output. How much of a problem this is depends on how frequently the hyper-network needs to be invoked. If only a few instances (and thus latent codes) are seen per batch, the hyper-network cost will be small. In cases like Rebain et al. (2022) however, where thousands of latent codes per batch are required, this cost can become prohibitive.

A related approach to hyper-networks is *feature-wise linear modulation* (FiLM) (Perez et al., 2018; Mehta et al., 2021; Chan et al., 2021), which similarly incorporates a secondary network to operate on latent codes, but restricts the output of this network to feature-wise *modulation* of activations in the MLP, rather than predicting all of its weights. This reduces the computational load of the hyper-network, but also reduces its expressiveness in terms of how it can affect the main MLP.

**Attention** − **Figure 4 (right)**. More recently, a class of neural field decoder architectures has emerged based on the sequence decoder mechanism used in transformers (Vaswani et al., 2017). Introduced by Jiang et al. (2021) and extended to higher-dimensional fields by Sajjadi et al. (2022), this architecture defines

a neural field using *cross-attention* layers. These layers compute attention between a query derived from the encoded sample position $\gamma(\mathbf{x})$, and a *set of tokens*, which fill a role equivalent to latent codes in the previously described architectures. This method of incorporating conditioning information from a *set-latent* representation (which subdivides the latent code into fixed-width tokens) provides unique benefits compared to other approaches, namely the ability to efficiently condition on very high-dimensional inputs and the ability to vary this conditioning in a position-dependent way. This efficiency comes from the fact that a set of latent tokens can encode information equivalent to the weights of a linear layer predicted by a hyper-network, but are produced by a network that is *shared over the tokens* rather than predicting the entire token set in a single operation. To quantify this, the complexity of predicting a set-latent is $O(n^2)$, compared to the $O(n^3)$ required to predict weights for a linear layer with a hyper-network[3]. Compared to other more efficient hyper-network-like architectures like FiLM, this also allows the flow of conditioning information into the coordinate network to vary over space, as the attention is derived from the encoded position. While it might seem at first glance that such set-latent prediction is not necessary for networks like auto-decoders, we have observed that the inclusion of a token-level mapping network can prevent training from collapsing when conditioning with a learned latent table.

A number of recent works (Müller et al., 2022; Chan et al., 2022; Liu et al., 2020; Chen et al., 2022) construct neural field style networks which condition on sample positions, but also incorporate a *spatial grid*, where each cell is associated with latent vector(s). Conceptually, this is similar to attention-based conditioning, as it also derives a sample-level feature from a larger set of vectors using weights derived from the sample position. Unlike the attention-based architectures we focus on, grid-based methods can perform spatial "look-up" nearly for free, significantly increasing their computational efficiency. Nonetheless, the application of these models to category-level models has been limited so far, and they suffer from the *curse of dimensionality* in scaling to problems beyond 3 dimensions. As such, we consider grids to be a complementary approach, and focus our analysis on the other architectures previously mentioned.

## 4 Experiments

Given the architectures described in Section 3.3, we are interested in analysing how they scale to very high-dimensional latent codes. In Section 4.1, we verify our assumptions on an *image auto-encoding* problem to investigate how changes in latent dimensionality and complexity of the data distribution affects performance. Then, in Section 4.2, we experiment on different application domains, each consisting of several datasets, to evaluate the latent scaling performance in practical settings. Finally, in Section 4.3, we test alternative concatenation schemes to what we choose for MLPs to demonstrate that we are performing a fair comparison against conditioning-by-concatenation.

### 4.1 Image auto-encoding

We implement a simple image auto-encoder which compresses images with a CNN to a latent representation of some dimension, using a neural field decoder to reconstruct the images from this latent code by interpreting it as a single latent code (concatenation, hyper-network) or a set-latent (attention). In all of these experiments we change only two variables for each dataset: *latent dimension* and *decoder architecture*. This allows us to isolate the variance between decoder architectures in their ability to utilize the information from the encoder. We test on two datasets, respectively discussed in Section 4.1.1 and Section 4.1.2, with network implementation details detailed in Section A.3.

### 4.1.1 Tiled MNIST – Figure 5 and Table 1

We design a dataset with controllable complexity to demonstrate how much the performance of an architecture can be affected by the size of the latent code and dimensionality of the data manifold. Loosely inspired by Lin et al. (2018), it consists of images formed by a $16{\times}16$ grid of MNIST digits where each digit is down-scaled to $16{\times}16$ pixels for a total image resolution of $256{\times}256$. The digits are chosen *randomly* from the $60,000$ images of the MNIST dataset, creating up to $60,000^{16\times16}$ unique possible combinations. This *on-the-fly*

---

[3]For simplicity, we assume that all network widths, token sizes, and token counts are equal to $n$

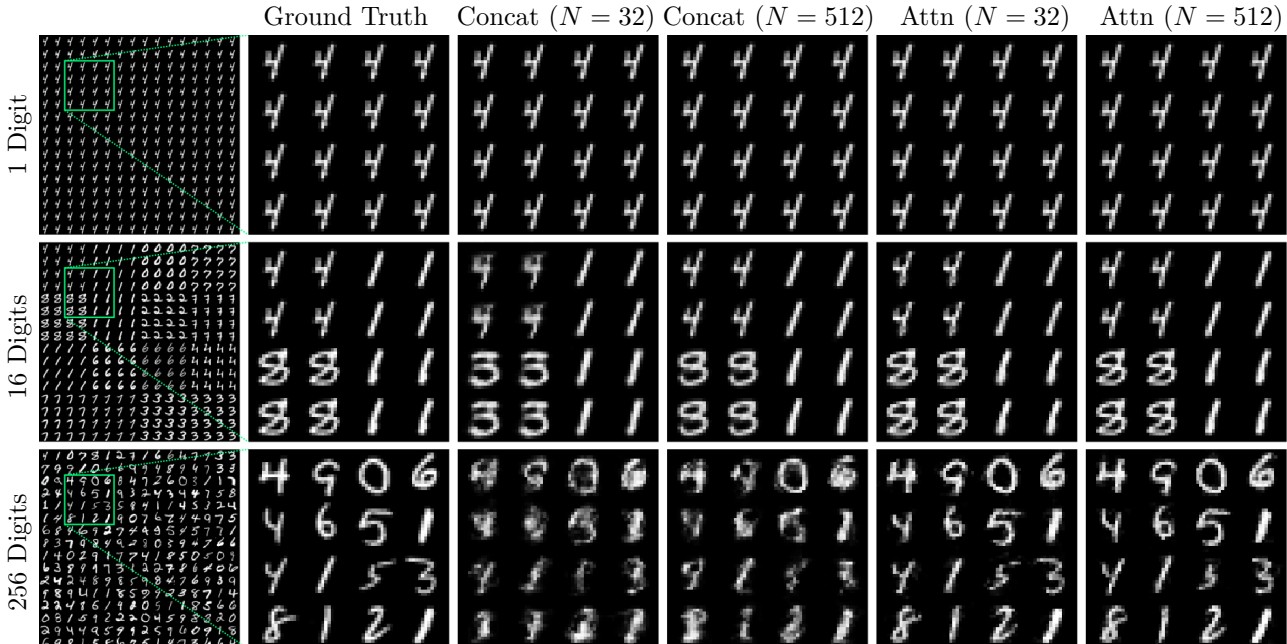

Figure 5: **Tiled MNIST** – Qualitative reconstruction results.

| # unique digits | 1 | | | 16 | | | 256 | | |
|---|---|---|---|---|---|---|---|---|---|
| Latent dimension ($N$) | 32 | 128 | 512 | 32 | 128 | 512 | 32 | 128 | 512 |
| Concatenation | 31.5 | 32.8 | 32.6 | 19.5 | 24.4 | 25.0 | 16.9 | 17.0 | 16.8 |
| Hyper-network | 30.8 | 33.3 | 33.7 | 19.3 | 24.4 | 25.4 | 18.8 | 21.7 | 21.8 |
| Attention | 35.0 | **36.3** | 33.9 | 26.1 | 31.3 | **31.4** | 23.5 | 24.3 | **24.6** |

Table 1: **Tiled MNIST** – We compare the reconstruction quality in terms of PSNR of three variants of our toy dataset. We find that the reconstruction quality drops significantly with data complexity, as expected, and that the best performing configurations for the two complex distributions are the attention-based model with the largest latent code. As expected, we also find that large latent dimensions are not helpful for very simple data.

image formation strategy ensures that memorization of the entire dataset is impossible. To test the effect of dataset complexity on network performance, we vary the number of *unique* digits in each image: either 1, 16, or 256, by setting consecutive 16×16, 4×4, or 1×1 blocks of digits to use a single repeated digit. We do this so that the images of each variant have the *same* size and visual properties, but vary *significantly* in the amount of information they contain.

**Analysis**. In Table 1, we report average reconstruction PSNR, leading to the following key observations:

- despite the similarity of the images, the reconstruction performance varies significantly between the variants as the number of unique digits increases, confirming that the more complex datasets are indeed more difficult to model;
- increasing latent code size correlates with reconstruction quality for the more complex datasets, but not for the single-digit dataset, confirming that latent code size affects reconstruction by accounting for the dimensionality of the data manifold;
- for concatenation-based decoders on the 256-digit variant, the quality is very poor and does not improve with latent code size; this suggests that the concatenation-conditioned network does not have the capacity to model this complex, high-dimensional data manifold.

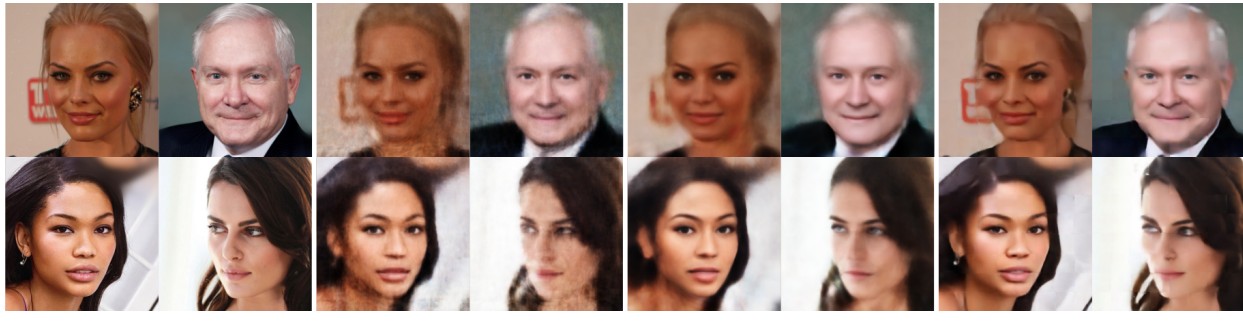

| Ground Truth | Concatenation | Hyper-network | Attention |

Figure 6: **Image auto-encoding** – Qualitative reconstruction results on CelebA-HQ for $N = 8192$.

| Latent dimension (N) | 512 | | | 2048 | | | 8192 | | |
|---|---|---|---|---|---|---|---|---|---|
| Decoder architecture | Concat | Hyper | Attn | Concat | Hyper | Attn | Concat | Hyper | Attn |
| CelebA-HQ | **21.4** | 21.1 | 21.3 | 24.7 | 23.5 | **26.0** | 24.5 | 23.6 | **26.3** |
| Decoder params ($\times 10^6$) | 0.6 | 2.1 | 1.7 | 1.0 | 2.2 | 1.7 | 2.6 | 2.5 | 1.7 |

Table 2: **Image auto-encoding** – We repeat our auto-encoding experiments on the CelebA-HQ dataset, and report the reconstruction quality (PSNR). We again find that the attention-based network in combination with high-dimensional latent codes provides the best performance. This advantage can not be explained by network capacity alone, as we find that the attention model performs the best for the largest codes, despite having the smallest number of parameters.

### 4.1.2 CelebA-HQ

In addition to the Tiled MNIST dataset, we also experiment with the CelebA-HQ dataset introduced by Karras et al. (2018). In contrast to the previous dataset, these are 30k real images consisting of high-resolution versions of CelebA (Liu et al., 2015) human face images. We resize these images to 256×256 resolution to keep the CNN encoder the same as for Tiled MNIST, and to limit the computational cost of training. In Table 2, we observe that attention-based decoder achieves the highest reconstruction PSNR, whereas concatenation and hyper-network decoders encounter saturation in performance earlier.

### 4.2 Novel view synthesis

For the remaining experiments, we focus on the more challenging real-world task of *novel view synthesis*, one of the main application domains of neural fields. Given one or more images of an object or a scene, this is the task of generating images from novel viewpoints. We experiment with two different neural field-based approaches to novel view synthesis: neural radiance fields (Mildenhall et al., 2020), and light field networks (Sitzmann et al., 2021). Both are analyzed using the following datasets, where we randomly select 10% of views to be held-out from training and used for testing:

- **HUMBI (Yu et al., 2020)**: is a large multiview dataset of 772 human subjects across a variety of demographics, captured with 107 synchronized HD cameras. We down-scale center crops of 97 views to 256×256 for 308 full body subjects. We multiplied the RGB values against a computed segmentation mask, such that the networks only reconstruct the subjects and not the background.
- **SRN Cars and Chairs (Sitzmann et al., 2019)**: We use the rendered dataset of cars and chairs from ShapeNet (Chang et al., 2015). The chair dataset consists of 3654 models with each instance being rendered from 200 unique viewpoints which we downscale to 128×128 resolution, while the cars dataset consists of 2151 models, with each instance being rendered from 250 unique viewpoints which we upscale to 256×256 resolution.

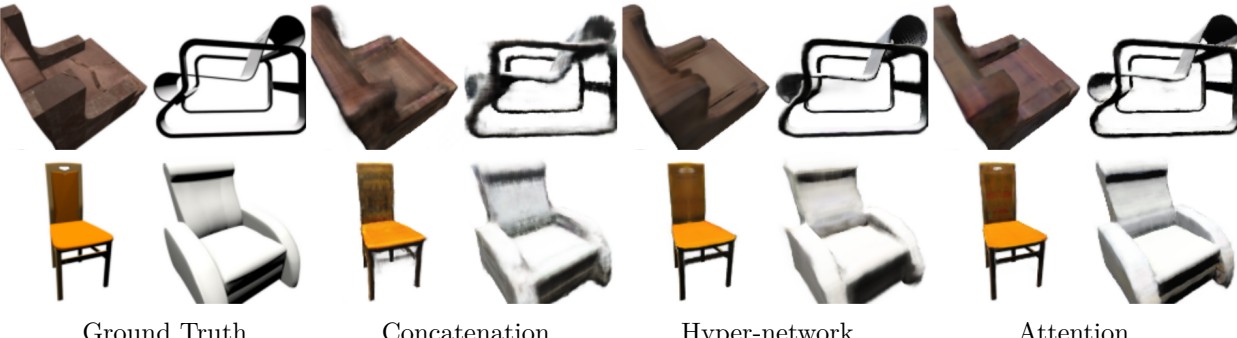

| Ground Truth | Concatenation | Hyper-network | Attention |

Figure 7: **Neural Radiance Fields** – Qualitative reconstruction results on SRN Chairs for $N = 8192$.

| Latent dimension ($N$) | 512 | | | 2048 | | | 8192 | | |
|---|---|---|---|---|---|---|---|---|---|
| Decoder architecture | Concat | Hyper | Attn | Concat | Hyper | Attn | Concat | Hyper | Attn |
| HUMBI | 29.3 | 30.2 | **31.0** | 27.9 | 30.6 | **31.0** | 28.3 | 29.9 | **31.0** |
| SRN-Chairs | 20.1 | **24.5** | 24.1 | 20.2 | 23.2 | **24.4** | 20.9 | 23.5 | **25.0** |
| SRN-Cars | 25.8 | 28.6 | **29.3** | 25.0 | 28.0 | **29.1** | 25.9 | 28.0 | **29.4** |
| Decoder params ($\times 10^6$) | 0.6 | 2.1 | 2.6 | 1.1 | 2.2 | 2.6 | 2.6 | 2.8 | 2.6 |

Table 3: **Neural Radiance Fields** – We compare the performance of different conditioning approaches, on the NeRF novel view synthesis task using the reconstruction PSNR metric on HUMBI, SRN-Chairs and SRN-Cars datasets. Attention conditioning with the largest latent size achieves the best result in all cases.

**Latent codes**. To avoid the complexity of deriving the parameters of a 3D scene from images through an encoder network, we adopt the approach of Rebain et al. (2022) for our novel view synthesis experiments, and implement an auto-decoder with a learnable latent table. The rows of the table, which each correspond to an object in the dataset, are interpreted as either single latent vectors, or set-latents with a fixed token width of 128 and varying number of tokens (such that the overall dimension is unchanged), depending on the decoder architecture.

**Training**. All novel view synthesis methods are supervised using the pixel-wise reconstruction loss in (2) applied on the training images and rendered pixel values for training views. For all datasets and architectures, training batches consist of 64 instances, with 2 views per instance, and 64 pixels sampled per image.

### 4.2.1 Neural radiance fields

A neural radiance field network $f_\theta$ defines a function from a point in space $\mathbf{x}$ and viewing direction $\mathbf{d}$ to radiance and optical density: $f_\theta : \mathbb{R}^5 \to \mathbb{R}^4$, $\mathbf{x} \mapsto f_\theta(\mathbf{x}, \mathbf{d}) = (\mathbf{c}, \sigma)$. The radiance and density values can be integrated along a ray using the approach described by Mildenhall et al. (2020) to produce rendered per-pixel color values. In our experiments, for simplicity, and because view-dependent effects are absent from two of our datasets and insignificant in the third, we omit the view direction inputs from the NeRF network. For volume rendering, we use the hierarchical sampling approach described by Mildenhall et al. (2020), but without allocating separate "coarse" and "fine" networks – instead sampling both coarse and fine values from the same network; specifically, we use 64 fine/importance samples, and 128 coarse/uniform samples for each ray, which we found to be the minimal counts required to avoid noticeable artifacts with our data.

**Analysis**. We report the reconstruction metrics on our test images in Table 3, and Figure 7 compares images rendered using different NeRF architectures. As these numbers show, attention-based decoders consistently achieve the best results for large latent codes, as well as the best performance overall.

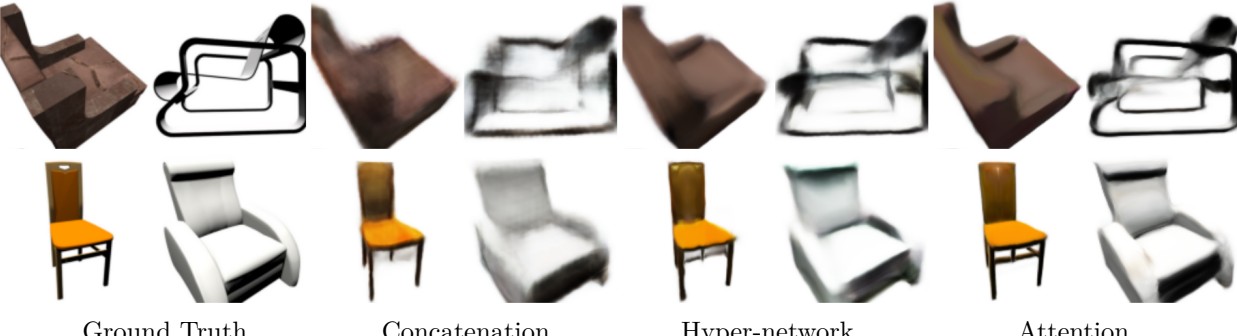

|     |     |     |     |
| :-: | :-: | :-: | :-: |
| Ground Truth | Concatenation | Hyper-network | Attention |

Figure 8: **Light Fields** – Qualitative reconstruction results on SRN Chairs for $N = 8192$.

| Latent dimension ($N$) | 512 | | | 2048 | | | 8192 | | |
| --- | --- | --- | --- | --- | --- | --- | --- | --- | --- |
| Decoder architecture | Concat | Hyper | Attn | Concat | Hyper | Attn | Concat | Hyper | Attn |
| HUMBI | 25.2 | 29.2 | **29.8** | 24.7 | 29.0 | **29.8** | 24.9 | 29.0 | **29.9** |
| SRN-Chair | 17.4 | 20.3 | **20.7** | 17.6 | 19.9 | **21.7** | 18.2 | 19.9 | **22.2** |
| SRN-Car | 23.8 | 26.8 | **27.5** | 23.8 | 26.3 | **28.1** | 24.2 | 26.3 | **28.4** |
| Decoder params ($\times 10^6$) | 0.6 | 2.2 | 2.6 | 1.1 | 2.3 | 2.6 | 2.7 | 2.9 | 2.6 |

Table 4: **Light Fields** – We compare the performance of different conditioning approaches, on the light field novel view synthesis task using the reconstruction PSNR metric on HUMBI, SRN-Chairs and SRN-Cars datasets. Only attention conditioning shows consistent improvement in PSNR on all datasets as latent size is increased.

### 4.2.2 Light field networks

Neural light fields define a function from ray to radiance: $f_\theta : \mathbb{R}^5 \to \mathbb{R}^3$, $\mathbf{x} \mapsto f_\theta(\mathbf{x}) = \hat{\mathbf{c}}$. In this formulation, the camera parameters map pixel coordinates directly to rays, which the light field network maps directly to pixel color values. For our experiments we employ Plücker coordinates (Sitzmann et al., 2021), which assign 5D ray coordinates to points on a 4D manifold, such that co-linear rays with the same direction map to the same point. This simplifies the mapping that the network needs to learn, though at the cost of requiring view rays to originate from a point outside the bounds of the scene.

**Analysis**. We report the reconstruction metrics on test images in Table 4, and qualitative results in Figure 8. In all cases, the transformer decoder achieves the highest reconstruction quality, and either increases with latent code size (SRN chairs and cars), or remains approximately constant (HUMBI) – indicating that either the data manifold dimensionality has been reached, or that the manifold is to complex for the network to model with the capacity it has. Hyper-networks remain approximately flat or decrease slightly with increasing latent size, indicating poor ability to integrate large conditioning vectors. Concatenation provides the worst performance on average, and show inconsistent behaviour with respect to latent size.

### 4.3 Ablation of concatenation methods

As our primary goal is to compare attention and concatenation as conditioning methods, it is important to verify that the approach we take to concatenating large latent codes into an MLP network is optimal. To this end, we perform an ablation study for different concatenation schemes using the largest latent code size of 8192 for one of our main experiment setups: CelebA-HQ auto-encoding.

| Variants | PSNR |
| --- | --- |
| 8-way split | **24.46** |
| 4-way split | 23.27 |
| 2-way split | 22.40 |
| 1 skip connection | 22.12 |
| no split or skip connection | 20.53 |

The first scheme we test is simply concatenating the entire latent
code to the first layer input. Next, we try adding a skip connection
from the input of this first layer to a layer half-way through the
network. For the rest, we try splitting the latent code evenly into
2, 4, or 8 parts, and concatenating these sub-vectors to different layers spread evenly through the network.
Note that the number of splits is limited by the depth of the network (8 layers in all our experiments). The
result of this study is shown in the inset table.

Splitting the latent code into 8 parts (as we do in our main experiments), and concatenating them to individual
layers provides the best results. This would suggest that the linear dimensionality reduction effect described
in Section 3.3 is indeed the limiting factor for concatenation efficiency. As such, we split all of our latent
codes for concatenation-based decoders to either 8 sub-codes with widths larger than the hidden layer width
of 256, or to $\lceil \frac{N}{256} \rceil$ sub-codes when the latent dimension $N$ is less than 2048.

## 5    Conclusion

In this paper we have provided an analysis of performance for some common conditioning architectures for
neural fields. We found a strong trend in favor of attention-based architectures being the most effective
option, particularly for cases where the conditioning signal is very high-dimensional. This result is valuable for
making decisions about architecture design for methods which require high-dimensional conditioning to model
complex data distributions. By sharing the results of these *very* computationally expensive experiments, we
hope to reduce the cost burden for future work in this area.

**Limitations (of our analysis)**. For architectures like hyper-networks which split computation into an
instance-level and sample-level stages, there is an inherent trade-off between the efficiency of the two stages.
For applications where new instances are seen infrequently, it can be advantageous to save resources by
having a more expensive instance-level computation and a cheaper sample-level one. If the application
requires many invocations of the instance-level network however, e.g. because training batches require many
hundreds or thousands of instances as in Rebain et al. (2022); Sajjadi et al. (2022), such a configuration
may be prohibitively inefficient. As detailed in Section A.1, we balance the hyper-paramater choices in our
networks such that each is comparable in efficiency when considering *both* the instance-level and sample-level
computations. This provides an analysis that is widely applicable to different applications, but may overlook
some application-specific routes for optimization that take advantage of caching the results of the instance-level
network.

**Broader impact**. Our analysis is fundamental research having broad implications for a wide range of ML
applications, and thus it is difficult, if not impossible, to anticipate all specific consequences of our research.
We do however note a few potential areas. First, we note that conditioned neural fields could be used for
generative methods, and thus come with all the consequent baggage of synthetic media generation. We feel
these issues have been widely discussed in the community at large, with our analysis not adding anything
new other than the potential for incrementally higher quality synthetic data.

Secondly, we note the large amount of energy required to run these experiments. We undertook this
investigation as a service to the community so that others don't have to, and make our source code available
for verifiability. Our results should allow others to make informed design decisions without needing to repeat
our experiments.

**Acknowledgements**. This work was supported by the Natural Sciences and Engineering Research Council of
Canada (NSERC) Discovery Grant, NSERC Collaborative Research and Development Grant, Google Research
(Canada), Digital Research Alliance of Canada, and Advanced Research Computing at the University of
British Columbia. We thank Wei Jiang, Weiwei Sun, Eric Hedlin, Daniel Watson, and David Fleet for their
comments.

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

# A    Appendix

## A.1    Balancing instance-level w/ sample-level computation

As previously noted, depending on workload, a conditional network may require more computation at the sample level (e.g. rendering all the pixels in a large image), or at the instance-level (e.g. batch sizes of hundreds or thousands of images). This can have a drastic effect on what architecture is more efficient – if most of the computation can be off-loaded from the sample level to the instance level, as in a large hyper-network, then the training or inference process may be much more efficient for cases where only few instances need to be considered at once. We chose to balance our choice of hyper-parameters for the case where such an efficiency gain is limited as is the case for methods like SRT and LOLNeRF. In practice for our problem domains, this means choosing batch sizes with large numbers of images with pixels sparsely sampled from each. The most noticeable effect of this choice is that we choose a network width for our hyper-networks of 64 neurons, as this keeps the memory and compute cost close to that of attention and concatenation for our training setup.

This results in most of our decoders having similar parameter counts, as the total number of flops is close to proportional to parameters when instance-level computations are not heavily re-used.

## A.2    Neural Field decoder architecture details

The decoder takes input coordinates $\mathbf{x}$ as input along with encoded latents $\mathbf{z}$. All networks use sinusoidal positional encoding to map raw coordinates to high-dimensional vectors before conditioning on them. For the concatenation and hyper-network decoders, the set-latent output of the encoder is flattened to a vector, while the set structure is retained for the cross-attention layers of the attention decoder. The network is supervised using the pixel-wise reconstruction loss of Eq. 2. The concatenation and hyper-network models both consist of 8-layer MLPs in all cases, while the attention models use 5 attention stages with three dense layers after each. With the exception of the hyper-network layers, all MLPs use a network width of 256 neurons. All multi-head attention layers use 16 heads and 256-dimensional keys unless otherwise specified.

## A.3    Image auto-encoding architecture details

We use an SRT-style (Sajjadi et al., 2022) network to encode an input image into patch-wise latent vectors; a CNN feature extractor maps the image to a $M \times M \times D$ feature map, from which $P$ patches are extracted and re-shaped to $P \times F$ latent tokens. These tokens are further processed with a single multi-head self-attention layer, resulting in $P \times G$ latent tokens $\mathbf{z}$ such that $PG = N$ (the latent dimension). In all cases, the encoder architecture is identical for each decoder architecture used.

The decoder takes pixel coordinates as input and directly outputs pixel color. In order to vary the latent dimension output from the encoder (equivalent here to the bottleneck width of the auto-encoder), the width and height of patches extracted from the last layer of the CNN before self-attention layers is varied such that the number of latent tokens varies while their individual dimensionality remains constant. We also configure the attention decoder for this application to use *single-head* attention (and correspondingly reduced the key dimension to 64), rather than multi-head as with the others, as we found it to provide slightly superior performance. This is likely due to the one-to-one nature of the mapping from output pixel coordinates to relevant information in the input features.

For training auto-encoders, we use a batch size of 128 images with 512 pixels per image.

## A.4    NeRF and light field auto-decoder architecture details

We directly store the latent code/set-latent values in a learnable latent table (initialized to zero), which is trained alongside the decoder network. For both hyper-network and attention models we include a learnable 64-dimensional embedding for each latent token which is initialized to a random value, sampled from a normal distribution – we found this to be necessary, as the initially zero-valued codes would otherwise cause the training of the layers conditioned only on latents to collapse. This embedding, unlike the latent codes, is

the same for each object in the dataset, so the non-zero initialization is not harmful to convergence. For concatenation, the latents are directly combined with previous layer activations, so we did not find additional embeddings to be necessary.

The NeRF decoder takes in 3D world-space sample positions as inputs, and outputs radiance and density values which are used to predict color for each ray according to the volume rendering equation described by Mildenhall et al. (2020). The light field decoder takes ray coordinates, mapped into Plücker coordinates, as input, and directly outputs pixel color for the ray.

For training auto-decoders, we use a batch size of 128 images with 64 pixels per image.

## A.5 Neural architecture details

In this section we give additional details on our neural network architecture choices. All MLPs are relu-activated, and use the original layer normalization strategy of the method each architecture is based on:

- Concatenation: none (Rebain et al., 2022)
- Hyper-networks: at each layer (Sitzmann et al., 2019)
- Attention: after skip connections (Sajjadi et al., 2022)

The network stage/layer count hyper-parameters were chosen to provide as close as possible to equal parameter counts across all experiments for the largest codes without changing the architecture for each experiment and while minimizing the number of hyper-parameters that are different from those used in previous works.

Finally, we give details specific to each conditioning method.

**Concatenation**. The concatenation decoder consists of an 8-layer MLP $f(x)$, which maps positional-encoded coordinates $x$ to the task specific output. Conditioning is achieved by splitting the latent code $z$ to 8 equally sized sub-codes and concatenating these to the input of each layer.

**Hyper-network**. The hyper-network decoder consists of the hyper-network $H(z)$, which maps latent codes $z$ to the weights of an 8-layer MLP $f(x)$, which maps positional-encoded coordinates $x$ to the task specific output. More specifically, the hyper-network first splits the latent code into 8 components, as in the concatenation-based decoder, then each of these sub-codes is then mapped by a 2-layer MLP to a weight and bias tensor to form one of the layers in $f(x)$.

**Attention**. The attention decoder is a function $g(x)$ which maps positional-encoded coordinates x to the task specific output. $g(x)$ consists of 5 network stages, each of which has a cross-attention layer followed by a 3-layer MLP, both of which are bridged by a skip-connection as in Scene Representation Transformer (Sajjadi et al., 2022). The cross-attention layer uses the same standard multi-head formulation as SRT and maps latent tokens to key and value vectors, and the input network activations from previous stages to query vectors. 16 heads are used for NeRF and light fields, and one head is used for image auto-encoding (as this gave slightly better results).

