# OpenReview forum: "Attention Beats Concatenation for Conditioning Neural Fields"
_TMLR — Accepted by TMLR_

### Review · Reviewer_NJ11 · 2022-10-05

**Summary Of Contributions:**

This paper studies the design choice of neural field conditioning. The author mainly considered 'concatenation', 'hyper-network', and 'attention' approaches in 2D, 3D, and 4D signals, where the backbone neural field is based on MLP- architecture with positional encoding. The authors empirically demonstrate that the 'attention' achieves the best performance across modalities and the scaling, i.e., enlarging the conditioning dimension, on the 'attention' strategy also shows consistent improvement.

**Requested Changes:**

**Mathematical Definition**: Some explicit mathematical definitions of the design choice (in the main text) will be encouraged.

**Intuition**: Providing some intuitions about the observation, i.e., "Attention" is better than others, will be great.

**more verification** (optional): Since this paper is an observational study, more extensive verification will be encouraged to convince the claim. While this is optional, more extensive experiments in the following area will truly make the paper stronger.
- (i) hyper-network design choices [1,2,3]
- (ii) neural field architectures [4,5]
- (iii) more modalities, e.g., audio, video

**Strengths And Weaknesses:**

**Strengths**

The observational study is very meaningful as the current neural field does not have a consensus on which design choice to use.

Experiments are considered over various neural field applications, e.g., image, 3d radiance field, and light field

------------

**Weakness**


The writing of the methodology part is slightly confusing.
- Except for the "Concatenation", the mathematical definition is not clear (as concatenation is somewhat trivial).
- Can the author provide the explicit mathematical definition of "Hypernetwork" and "Attention"?


While the finding is very interesting, it is slightly unclear why "Attention" is the best choice.
- Can the author provide some intuition about the observation?


While authors have roughly categorized the design choices (into three), there exist various methods that are known to be effective. Considering more design choices will definitely give more intuition to the readers.
- Functa [1]: only adds a bias term on a globally shared network
- CIPS [2]: similar to Pi-GAN
- Modulated SIREN [3]


More extensive experiments across neural field architectures
- For instance, does this observation holds for various neural field architectures such as SIREN [4], and FFN [5]?

------------

**Questions**\
Have the author tried such observational studies on different modalities such as audio (t) or video (x,y,t)

------------

**Reference**\
[1] Dupont et al., From data to functa: Your data point is a function and you can treat it like one, ICML 2022\
[2] Anokhin et al., Image Generators with Conditionally-Independent Pixel Synthesis, CVPR 2021\
[3] Mehta et al., Modulated Periodic Activations for Generalizable Local Functional Representations, ICCV 2021\
[4] Sitzmann et al., Implicit Neural Representations with Periodic Activation Functions, NeurIPS 2020\
[5] Tancik et al., Fourier Features Let Networks Learn High Frequency Functions in Low Dimensional Domains, NeurIPS 2020

---

> ### Author Response · Authors · 2022-10-19
> **Response to Reviewer NJ11**
>
> Thank you for your detailed feedback. We will add more specific definitions of hyper-networks and attention in a revision. We will also publish our source code upon acceptance.
>
>  > Can the author provide some intuition about the observation?
>
> Intuitively, there seem to be two major factors that likely affect this:
>  - compared to linear dimensionality reduction (which is essentially what concatenation and hypernetworks do), cross-attention layers provide a more computationally efficient way of extracting a feature vector from the latent code;
>  - attention enables different "parts" of the latent code to easily be mapped to different areas of the input space, allowing for simple learning of spatially decomposed representations.
>
>  > While authors have roughly categorized the design choices (into three), there exist various methods that are known to be effective. Considering more design choices will definitely give more intuition to the readers.
>
> We are limited by the computational resources that our experiments require to train with high dimensional inputs on very large datasets. For context, our jobs (corresponding to individual table entries) take 400-1000 GPU-hours each to train on V100s, which corresponds to a cost of 1000-2500 USD each at current GCP prices. We have carefully chosen our architectures to be comparable to each other, but also similar to prior work that has demonstrated good results for similar training schemes and datasets (namely Scene Representation Networks, CodeNeRF, LOLNeRF, and Scene Representation Transformers) in an effort to be maximally useful to future work that builds on this particular stream of research.
>
>  > Have the author tried such observational studies on different modalities such as audio (t) or video (x,y,t)
>
> We chose data modalities to align with prior work that could provide examples of problem domains where conditional neural fields have already been shown to be effective at modeling large datasets with high-dimensional conditioning. In this way we focus on our goal of optimizing architecture, rather than spending time exploring applications to unfamiliar data. As such, we leave the interesting question of how to best support additional modalities to future work.

---

### Review · Reviewer_8ztu · 2022-10-14

**Summary Of Contributions:**

This paper considers the topic of conditioning neural fields by a latent vector, and aims to find the best way to infuse that information into the network. The authors consider several popular choices, and run extensive experiments, comparing them across various datasets and latent vector sizes. They conclude that attention over a set of latent subvectors works best, especially at scale.

**Broader Impact Concerns:**

I see no unaddressed ethical concerns, and the discussion the authors include in their paper is sufficient.

**Requested Changes:**

I think the work is already in good shape, and I don't have significant changes to request.

**Strengths And Weaknesses:**

=== Strengths ===

- (S1): The authors pose a very concrete question, and do a thorough job at trying to answer it. They experiment across various datasets with differing complexities, ranging from toy to large scale. In the toy experiment with MNIST, the authors reduce the intrinsic dimensionality while retaining the image size by duplicating digits, which allows them to decouple these two factors; I found this idea quite clever.
- (S2): The writing is excellent: it reads very clearly, they are no typos/errors, and overall feels quite polished. The basics of what neural fields are and how they are used are well-explained. I also liked the inclusion of Figure 2 to visualize the dependencies between the different works.

=== Weaknesses ===

I did not observe significant weaknesses, given the type of the work. While clearly there is no novelty presented in the paper, this is expected from a "large scale evaluation"-style work, and it doesn't make it any less useful. I have some clarification questions (rather minor) which I defer to the "Questions" section below.

=== Questions ===
- (Q1): In the attention conditioning, the network queries the set-latent with a query based on the location x. Is it fair to assume that each of the vectors in the set-latent will possibly end up corresponding to a part of the object that is a coherent slice of the space? In other words, does it make sense to view this as a relaxed/learned version of having a different latent vector per every voxel of some predefined grid?
- (Q2): At the bottom of page 6, the authors talk about the complexity of predicting the set-latent and how that compares to the complexity of predicting the network weights using a hypernetwork. Is this discussion specific to the auto-encoder setup (as in the auto-decoder the latent would not have to be predicted)?

=== Nitpicks ===
- In "Our analysis is fundamental research (...), and thus difficult, if not impossible, to anticipate all specific consequences of our research", shouldn't there be a verb somewhere near "difficult" e.g. "it is difficult"?

---

> ### Author Response · Authors · 2022-10-19
> **Response to Reviewer 8ztu**
>
> Thank you for your thoughtful review and comments.
>
>  > (Q1): In the attention conditioning, the network queries the set-latent with a query based on the location x. Is it fair to assume that each of the vectors in the set-latent will possibly end up corresponding to a part of the object that is a coherent slice of the space? In other words, does it make sense to view this as a relaxed/learned version of having a different latent vector per every voxel of some predefined grid?
>
> Essentially, yes. The spatial attention mechanism is indeed similar to a feature field encoded in a voxel grid, though without the limitations on resolution, extent, and dimensionality imposed by a grid. Future work could explore hybrid models which combine explicit grids and set-latents as conditioning.
>
>  > (Q2): At the bottom of page 6, the authors talk about the complexity of predicting the set-latent and how that compares to the complexity of predicting the network weights using a hypernetwork. Is this discussion specific to the auto-encoder setup (as in the auto-decoder the latent would not have to be predicted)?
>
> This would not apply to any situation where no token-level mapping is required. However, we use such a mapping network in all cases, as performing attention directly on zero-initialized latents at the beginning of training in the auto-decoder caused training to collapse in our preliminary experiments. We will edit the section to make this clearer.

---

> > ### Comment · Reviewer_8ztu · 2022-11-03
> > **Response to authors**
> >
> > Thanks; your answers make sense.

---

### Review · Reviewer_HshV · 2022-10-17

**Summary Of Contributions:**

The authors conduct a large scale study on the architecture bias to condition neural fields. They compare concatenation, HyperNetworks and an Attention-based concatenation mechanism on different datasets and observe an advantage over most setups for the  Attention-based concatenation mechanism.

**Broader Impact Concerns:**

Does not apply I think.

**Requested Changes:**

Please provide a clear description of the architectures, their parameter count, their training time, hyperparameters used. Although you try to describe the different architecture in words, I am not sure what you do, until I see a proper mathematical description how exactly the 3 versions, especially the HyperNetwork and the Attention version, are constructed and what happens if you change e.g. the latent dimension.

In general, I am not convinced about the approach put forward in this paper. I would have liked a much more fine grained study + ablations on some moderate dataset, e.g. the MNIST one studied here, but where the architecture search is much more thorough. What roles do initialisation, regularisation (strength), learning rates, etc play.
Also architecture wise,  the HyperNetwork can also be used with sharing meaning that the HyperNetwork gets as input a "layer token" and a conditioning z token producing the weights of every single layer (see e.g. https://arxiv.org/abs/1906.00695)

In the ablation that you did, even more splits for the concatenation method then leads to even improved performance, better than the attention mechanism?
Also, please provide ablations for the attention mechanism. When does it fail, what makes it work?
It seems that often performance of methods decrease with more dimensionality, can you try different initialisation schemes to prevent that from happing? Or is this related to regularisation and one wants a lower dimensional latent space? Why does this not apply for the attention base mechanism?

More technical issues:

1. "If we wish to perfectly model some distribution of data, then we must use a latent dimension at least equal to that of the manifold on which that distribution lies." I would be very careful with these kind of statements. I agree that the latent dimension plays a role in practice to model a mapping from Z to some Image distribution but theoretically this is a much more delicate issue since one can densely sample from a high prob distribution through a mapping coming from a one-dim (e.g. Uniform) distribution, see e.g. https://arxiv.org/abs/2006.16664.
Since this is one of the major claims of the paper, I would study these issues in detail.

2. "For example, choosing ρ(·) = ∥ · ∥22 results in the codes Z being distributed as a Gaussian". I think this is not true: You are not modeling the posterior or a distribution as far as I understand but a set of parameters for which you penalize large values. Why would these "samples" converge to one from a gaussian?


**Strengths And Weaknesses:**

In general I find the details given in the manuscript not sufficient to 1) understand properly what they did 2) how the experiments were conducted. This makes is not possible to evaluate the study, see below.

---

> ### Author Response · Authors · 2022-10-19
> **Response to Reviewer HshV**
>
> Thank you for your comments and suggestions. We will prepare a revision to more clearly and concisely summarize the architectural and training details. Parameter counts for each architecture and latent size are already provided in the main experiment tables.
>
>  > Also architecture wise, the HyperNetwork can also be used with sharing meaning that the HyperNetwork gets as input a "layer token" and a conditioning z token producing the weights of every single layer (see e.g. https://arxiv.org/abs/1906.00695)
>
> This is true, but because our hypernetwork already contains a bottleneck between the latent input and each weight matrix output (as is required for the computation to remain feasible when scaling up the latent code size), this would decrease only the parameter count and not the computational cost.
>
>  > In the ablation that you did, even more splits for the concatenation method then leads to even improved performance, better than the attention mechanism?
>
> If you mean more than 8 splits, this is not possible, as each split is routed to one MLP layer, and we use only 8 layers.
>
>  > I would have liked a much more fine grained study + ablations on some moderate dataset, e.g. the MNIST one studied here, but where the architecture search is much more thorough. What roles do initialisation, regularisation (strength), learning rates, etc play.
>  ...
>  Also, please provide ablations for the attention mechanism. When does it fail, what makes it work? It seems that often performance of methods decrease with more dimensionality, can you try different initialisation schemes to prevent that from happing? Or is this related to regularization and one wants a lower dimensional latent space? Why does this not apply for the attention base mechanism?
>
> Could you elaborate more specifically on how you believe our current experiments fail to support our conclusions? While our Tiled MNIST dataset is a "toy" in the sense that it is artificial, it is not any cheaper to run experiments on than the others. Testing all possible parameters regardless of their relevance to our research question is neither feasible nor consistent with the TMLR guidelines, which require only experimental results that support our claims. If you believe there is a specific parameter we have not evaluated that you think is likely to affect the validity of our analysis, we would very much appreciate it if you could provide your reasoning for why that is the case.
>
>  > "If we wish to perfectly model some distribution of data, then we must use a latent dimension at least equal to that of the manifold on which that distribution lies." I would be very careful with these kind of statements. I agree that the latent dimension plays a role in practice to model a mapping from Z to some Image distribution but theoretically this is a much more delicate issue since one can densely sample from a high prob distribution through a mapping coming from a one-dim (e.g. Uniform) distribution, see e.g. https://arxiv.org/abs/2006.16664. Since this is one of the major claims of the paper, I would study these issues in detail.
>
> We are aware that such *approximations* exist, which is why we say "to perfectly model" here. Regardless, our focus is not on making theoretical claims about this relationship, but on the effects of it that we observe empirically. We will make this clearer in the text.
>
>  > "For example, choosing ρ(·) = ∥ · ∥22 results in the codes Z being distributed as a Gaussian". I think this is not true: You are not modeling the posterior or a distribution as far as I understand but a set of parameters for which you penalize large values. Why would these "samples" converge to one from a gaussian?
>
> This is not our claim, but rather was derived by Park et al. in DeepSDF (which we cite) and has been used in a number of works since then. Could you clarify whether you believe the original publication was incorrect or if you believe that we are applying it incorrectly?

---

### Author Response · Authors · 2022-10-31
**Revision**

We have uploaded a revision which includes the requested additional architecture details. As we have not received clarification on specific experiments that would be required to support our claims, we have not performed any additional experiments.

We again thank the reviewers for their helpful comments and suggestions.

---

### Decision · Action_Editors · 2022-11-14

**Recommendation:** Accept as is

**Comment:**

This paper studies the design choice of neural field conditioning, and considers 'concatenation', 'hyper-network', and 'attention' approaches in 2D, 3D, and 4D signals, where the backbone neural field is based on MLP architectures. The authors empirically demonstrate that the 'attention' achieves the best performance.

It received three reviews. After rebuttal, two reviewers recommended Accept, and one reviewer recommended Leaning Accept. On one hand, all the reviewers agree that the authors have posed a very concrete question, the empirical study is meaningful, and experiments are comprehensive. The paper is also well written. On the other hand, two reviewers mentioned that more clear definition of "HyperNet" and "Attention" is needed. Reviewer HshV suggested extra experiments to further strengthen the paper. In the paper revision, the authors have provided clarification on additional architecture details.

Overall, all the reviewers are happy about the paper, and the Action Editor would like to recommend accept of the paper.

**Audience:**

Yes, some individuals in TMLR's audience will be interested in this paper.

**Claims And Evidence:**

The claims are accurate and convincing with clear evidence.